# Pharmacist involved education program in a multidisciplinary team for oral mucositis: Its impact in head-and-neck cancer patients

Kensuke Yoshida[1,2]*, Yasumitsu Kodama[1☉¤a], Yusuke Tanaka[2☉], Kyongsun Pak[3☉¤b], Marie Soga[4‡], Akira Toyama[2‡], Kouji Katsura[4‡], Ritsuo Takagi[1]

1 Division of Oral and Maxillofacial Surgery, Faculty of Dentistry & Graduate School of Medical and Dental Sciences, Niigata University, Niigata, Japan, 2 Division of Hospital Pharmacy, Niigata University Medical and Dental Hospital, Niigata, Japan, 3 Division of Biostatistics, Center for Clinical Research, National Center for Child Health and Development, Tokyo, Japan, 4 Division of Oral Radiology, Niigata University Medical and Dental Hospital, Niigata, Japan

☉ These authors contributed equally to this work.
¤a Current address: Division of Oral and Maxillofacial Surgery, Niigata University, Niigata, Japan
¤b Current address: Division of Biostatistics, Center for Clinical Research, Tokyo, Japan
‡ These authors also contributed equally to this work.
* kensukeyoshida-nii@umin.ac.jp

**Data Availability Statement:** All relevant data are within the manuscript and its Supporting information files.

# Abstract

## Objectives

This retrospective study examined how a pharmacist-involved education program in a multidisciplinary team (PEMT) for oral mucositis (OM) affected head-and-neck cancer (HNC) patients receiving concurrent chemoradiotherapy (CCRT).

## Materials and methods

Total samples data of 53 patients during the stipulated timeframe were retrospectively collected from electronic medical records from February 2017 to January 2019. We compared the presence/absence of OM (OM: yes/no) between patients with and without PEMT (PEMT: yes/no) as the primary endpoint and OM severity as the secondary endpoint. The following information was surveyed: age, gender, weight loss, steroid or immunosuppressant use, hematological values (albumin, white blood cell count, blood platelets, and neutrophils), cancer grade, primary cancer site, type and use of mouthwash and moisturizer, opioid use (yes/no, days until the start of opioid use, and dose, switch to tape), and length of hospital day (LOD). The two groups were compared using Fisher's exact test for qualitative data and the Mann-Whitney U test for quantitative data, and a significance level of $p<0.05$ was set.

## Results

The group managed by PEMT had significantly lower weight loss and a significantly lower incidence of local anesthetic and opioid use and switch to tape compared with the group not managed by PEMT ($p<0.05$). The two groups showed no significant difference in OM (yes/

**Funding:** The authors received no specific funding for this work.

**Competing interests:** The authors have declared that no competing interests exist.

no) or OM severity. The PEMT group had significantly shorter LOD at 57 (53–64) days compared with the non-PEMT group at 63.5 (57–68) days ($p<0.05$).

## Conclusions

Our results showed that PEMT did not improve OM (yes/no) or OM severity in HNC patients undergoing CCRT. However, the PEMT group had a lower incidence of grades 3 and 4 OM than the non-PEMT group, although not significantly. In addition, PEMT contributed to oral pain relief and the lowering of the risk for OM by reduction in weight loss.

## Introduction

Oral mucositis (OM) has been described as the most painful aspect of chemotherapy [1]. Due to increased pain attributed to OM, the ingested amount of food is decreased, motivation with respect to treatment is reduced, and treatment-related deaths can occur, all of which warrant a change in the therapeutic approach [2]. The rate of OM has been reported to be 91% and 66% in patients with grades (Grs) 3 and 4 OM, respectively [3], receiving concurrent chemoradiotherapy (CCRT) for head and neck cancer (HNC).

The Multinational Association of Supportive Care in Cancer and European Society for Medical Oncology guidelines [4, 5] report that moisturizing agents are protective and stimulate brittle mucosa, and mouth cleaning can reduce the risk of mucosal infection by bacteria. Furthermore, the provision of oral care by dentists and dental hygienists has been reported to be highly effective in the treatment of OM, reducing its severity, shortening the length of hospital day (LOD), and reducing the dosage of morphine when it is required [6–9]. Patient education programs are also an important aspect of oral care [10]. Indeed, reports of randomized studies indicate that OM decreases as a result of educational interventions for patients with acute myeloblastic leukemia [11]. A multidisciplinary medical team approach has been shown to be protective against OM in patients. Such approach supports treatment and evaluations by dentists, professional oral care delivered by dental hygienists, and daily care and mental health follow-ups provided by nurses. In contrast, few reports describe the contributions made by pharmacists in a medical care team. In our hospital, pharmacists have been providing patient education on oral care since February 2018 with the goal of increasing adherence to mouth washing and moisturizing. These pharmacists were trained via an oral care program offered by dentists. In this study, we examined how a pharmacist-involved education program in a multidisciplinary team (PEMT) for OM affected HNC patients receiving concurrent chemoradiotherapy (CCRT).

## Materials and methods

### Subjects

This study was performed with the approval of the Research Ethics Niigata University Hospital (approval number 2018–0201). All data were fully anonymized, so the Research Ethics Niigata University Hospital waived the requirement for informed consent. The subjects were patients who were undergoing CCRT from February 2017 to January 2019. All patients received the first cycle of chemotherapy. Pharmacist was educated by a dentist in January 2018. Therefore, the surveillance period was chosen to compare the year prior to January 2018, when no education was received, and the year after January 2018, when education was received. All patients

received CCRT on a regimen of cisplatin 80 mg/m$^2$ on days 1, 22, and 43 during the aforementioned time period.

A retrospective survey of electronic medical records was conducted with the cooperation of the Medical Information Department of Niigata University Hospital. The patient data collected included age, gender, weight loss, steroid or immunosuppressant use, hematological values (albumin, white blood cell count, blood platelets, and neutrophils), cancer grade, primary cancer site, type and use of mouthwash and moisturizer, opioid use (yes/no, days until the start of opioid use, and dose), switch to fentanyl tape, and LOD. All patients who underwent CCRT during the investigation period were surveyed as a countermeasure against selection bias. Investigators were not involved CCRT as a countermeasure against medical surveillance bias. Presence/absence of OM and grade evaluator are not all author.

## Definitions of terms

Weight loss was defined as the difference in body weight before the initiation of CCRT (0 Gy) and after the delivery of all fractions (70 Gy). Hematological values were the median of four points (0, 20, 40, and 60 Gy). The presence or absence of OM and its severity were evaluated using an oral radiology criterion based on the Common Terminology Criteria for Adverse Events (CTCAE v5.0). The opioid dose used was determined from the oral morphine conversion ratio of the Japanese Society for Palliative Medicine as follows: oral morphine: oral oxycodone: transdermal fentanyl = (30:20:1) [12]. Patients who have already used opioids before the target period are not included summary of data for patients receiving opioids. The LOD was the period from the CCRT start date to discharge from the hospital.

## Mouthwash type

There were four types of mouthwash used: sodium azulene sulfonate hydrate (SASH) preparation, SASH + sodium bicarbonate (NaHCO$_3$) + lidocaine hydrochloride viscous, steroidal anti-inflammatory agent, and the Japanese traditional drug Hangeshashinto. The moisturizer used was dimethylisopropylazulene ointment.

## Pharmacist-delivered education program in the multidisciplinary team

Dentists provided an oral care program of a total duration of 240 minutes which was designed to train pharmacists in a didactic manner using a 64-slide PowerPoint presentation. The program delivered an overview of oral care (oral care need, OM mechanism, and OM-related adverse events), management (pain management, dental care, and general care), and oral assessment (graded by points using photographs of patients).

The pharmacists made medication instruction manual and explained in person to the patients at least once a week using a medication instruction manual (mechanism of drug, usage, and side effects). Listed below is a summary of the oral care program (Fig 1).

1. Need to continue using a mouthwash and a moisturizer
   The pharmacist explained to the patient as follows:
   If you do not use a mouthwash, your mouth cannot be kept clean.
   If you do not moisturize, the moisture will decrease in the lining inside your mouth.
   From the above, OM is likely to occur and to become more severe once it occurs.

2. Frequency of use of at least 6 times a day
   The pharmacist explained to the patient as follows:
   Make it a habit to use a mouthwash and moisturizer at least 6 times a day: when you wake up, after breakfast, after lunch, around 15:00, after dinner, and before going to bed. This

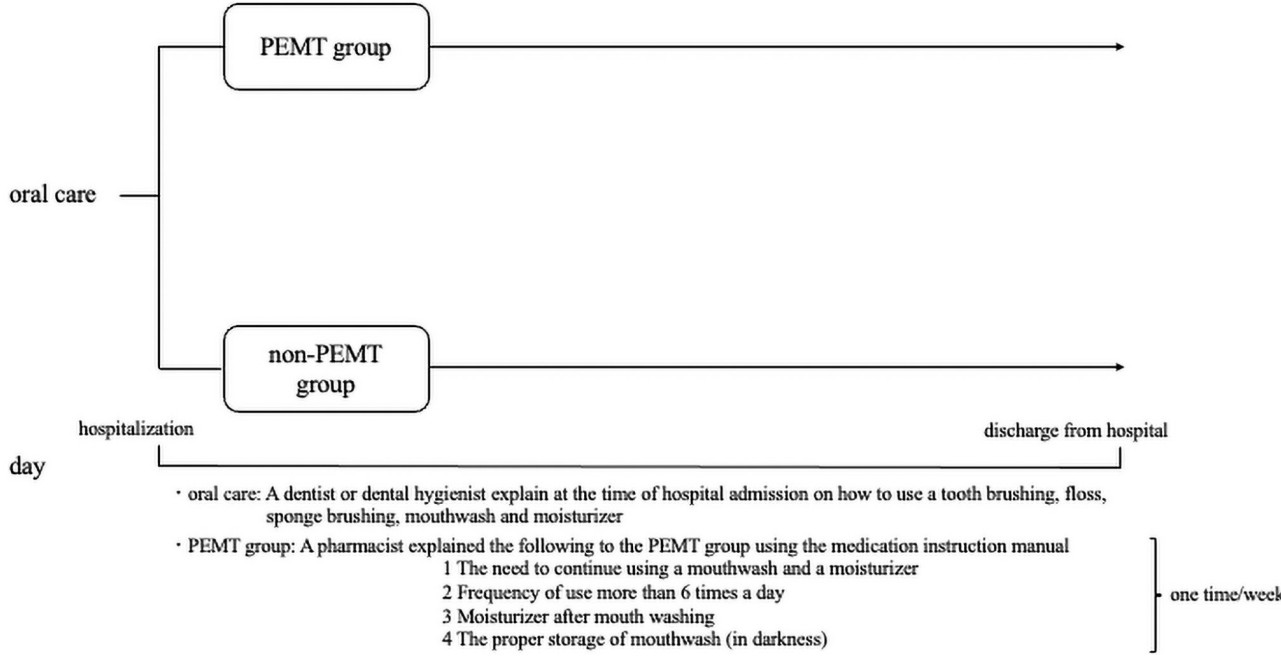

**Fig 1. Education program by pharmacist's involvement in the multidisciplinary team.**

regimen helps to maintain the cleanliness of the oral cavity and the amount of intraoral moisture.

3. Moisturizer after mouth washing
   The pharmacist explained to the patient as follows:
   If you use a moisturizer first, the effect of the mouthwash may be weakened.

4. Proper storage of mouthwash (in darkness)
   The pharmacist explained to the patient as follows:
   The mouthwash used was sodium azulene sulfonate hydrate (SASH). If SASH is placed in a bright place, the effect of SASH may be weakened.

## Statistical analysis

Patient background characteristics were summarized by group (PEMT: yes/no). The two groups were compared using Fisher's exact test for qualitative data and the Mann-Whitney U test for quantitative data, and a significance level of $p<0.05$ was set. JMP$^®$ 14.2 (SAS Institute Inc., Cary, NC, USA) was used for statistical analysis. It is an exploratory study, and there were no similar reports in the past, so sample size could not be calculated. Therefore, from the viewpoint of feasibility, we investigated two-year survey period and conducted a retrospective survey.

## Results

### Background

Of the 53 patients enrolled in this study, 23 patients received PEMT and 30 patients did not. The combination of SASH + sodium bicarbonate (NaHCO$_3$) + lidocaine hydrochloride viscous was used by 0% of the patients who received PEMT versus 30% (9 patients; $p<0.01$ Table 1) of those who did not.

**Table 1. Background characteristics by patient groups.**

| | Received PEMT | | |
|---|---|---|---|
| | **Yes** | **No** | **p** |
| | **(n = 23)** | **(n = 30)** | |
| Age | 64 (56–77) | 64 (59–69.8) | 0.47[b] |
| Gender (male:female) | 21:2 | 29:1 | 0.81[c] |
| Steroid use(%) | 1 (4.4%) | 0 | 0.89[c] |
| Immunosuppressant use | 0 | 0 | - |
| Hematological values | | | |
| Albumin (g/dL)[a] | 3.6(3.5–3.8) | 3.6(3.4–3.9) | 0.91[b] |
| WBC count (/μL)[a] | 4570 (3655–5705) | 4740 (4198.8–5377.5) | 0.73[b] |
| Neutrophils (/μL)[a] | 3345 (2565–4050) | 3550 (3097.5–3947.5) | 0.51[b] |
| Platelets (/μL)[a] | 18.1 (16.2–25.5) | 19.5 (16.4–23.1) | 0.88[b] |
| eGFR (mL/min/1.73m$^2$)[a] | 80.1 (69.2–86.2) | 80.1 (70.7–88.0) | 0.50[b] |
| Serum Cr (mg/dL)[a] | 0.8 (0.7–0.8) | 0.75 (0.7–0.9) | 0.66[b] |
| CRP (mg/dL)[a] | 0.2 (0.1–0.4) | 0.3 (0.1–0.7) | 0.30[b] |
| Cancer grade (%) | | | |
| 1 | 3 (13.0%) | 2 (6.9%) | 0.46[c] |
| 2 | 5 (21.7%) | 11 (36.7%) | |
| 3 | 6 (26.1%) | 10 (34.5%) | |
| 4 | 9 (39.1%) | 7 (24.1%) | |
| Primary cancer site | | | |
| Epipharynx | 2 | 5 | 0.85[b] |
| Oropharynx | 5 | 7 | |
| Hypopharynx | 9 | 9 | |
| Salivary gland | 2 | 3 | |
| Nasopharynx | 1 | 0 | |
| Maxillary sinus | 1 | 2 | |
| Larynx | 3 | 3 | |
| Tongue | 0 | 1 | |
| Type (%) | | | |
| Sodium azulene sulfonate hydrate preparation | 23 (100) | 29 (96.7) | 0.89[c] |
| Dimethylisopropylazulene ointment | 23 (100) | 29 (96.7) | 0.89[c] |
| Sodium azulene sulfonate hydrate + sodium bicarbonate + lidocaine hydrochloride viscos | 0 | 9 (30) | 0.01[c] |
| Steroidal anti-inflammatory agent | 0 | 3 (10) | 0.34[c] |
| Japanese traditional drug Hangeshashinto | 0 | 3 (10) | 0.34[c] |
| Dosage | | | |
| Sodium azulene sulfonate hydrate preparation (mL)[a] | 45 (25–80) | 62.5 (39–83) | 0.16[b] |
| Dimethylisopropylazulene ointment (g) [a] | 120 (60–180) | 120 (60–185) | 0.76[b] |

PEMT = pharmacist-involved education program in a multidisciplinary team.

eGFR = estimated glomerular filtration rate, Cr = creatinine, CRP = C-reactive protein, RT = radiotherapy.

[a] Data are the median (interquartile range(IQR)) unless otherwise indicated.

[b] Mann-Whitney U test.

[c] Fisher's exact test.

**Table 2. Weight loss.**

|  | Received PEMT | | |
| --- | --- | --- | --- |
|  | **Yes** | **No** | ***p*** |
|  | **(n = 23)** | **(n = 30)** |  |
| Weight loss (kg)[a] | 3.1 (1.3–4.9) | 4.3 (2.6–5.8) | 0.048[a] |

PEMT = pharmacist-involved education program in a multidisciplinary team.

[a] Data are the median (interquartile range(IQR)) unless otherwise indicated.

**Table 3. Summary of data for patients receiving opioids.**

|  | Received PEMT | | |
| --- | --- | --- | --- |
|  | **Yes[a]** | **No[b]** | ***p*** |
|  | **(n = 22)** | **(n = 25)** |  |
| Opioid use (%) | 15 (65.2) | 23 (92) | 0.09[c] |
| Days until opioid use median(IQR) | 19(17–21.5) | 18 (15–21.5) | 0.20[d] |
| Opioid dose (mg) median (IQR) | 1102.5 (0–1736.3) | 1860 (1050–2850) | 0.01[d] |
| Switching to fentanyl tape (%) | 5 (22.7) | 16 (64) | 0.01[c] |

PEMT = pharmacist-involved education program in a multidisciplinary team.

[a] One patient excluded before concurrent chemoradiotherapy (CCRT).

[b] Five patients excluded before CCRT.

a and b: Because patient(s) was/were on opioids before CCRT.

[c] Fisher's exact test.

[d] Mann-Whitney U test.

※ Oral morphine: oral oxycodone: transdermal fentanyl = (30:20:1).

**Weight loss.** Weight loss was lower in the group that received PEMT (*p<0.05*; Table 2).

**Opioids.** Opioids were used by 65.2% of patients (15 cases) who received PEMT and by 92% (23 cases) of those who did not; although the difference between groups was not significant (Table 3). Days until the start of opioid use were 19 (17–21.5) days in the PEMT group and 18 (15–21.5) days in the non-PEMT group, resulting in no significant difference between groups. The median opioid dose was significantly different between groups; 1102.5 (0–1736.3) mg in the PEMT group and 1860 (1050–2850) mg in the non-PEMT group (*p<0.01*). Oral oxycodone was switched to fentanyl tape in a significantly lower percentage of patients in the PEMT group (22.7%; 5 cases) than non-PEMT group (64%; 16 cases) (Table 3).

**Severity of OM.** The severity of OM by CTCAE grade is depicted for the PEMT group and non-PEMT group (PEMT group: Gr 0, 26.1% of patients [6 cases]; Gr 1, 13.0% [3 cases]; Gr 2, 52.2% [12 cases]; Gr 3, 8.7% [2 cases]; and Gr 4, 0% [0 cases]; non-PEMT group: Gr 0, 13.3% [4 cases]; Gr 1, 6.7% [2 cases]; Gr 2, 63.3% [19 cases]; Gr 3, 13.3% [4 cases]; and Gr 4, 3.3% [1 case]). There was no significant difference between the groups (Fig 2).

**Length of hospital days.** The PEMT group had significantly fewer days of LOD at 57 (53–64) days compared with the non-PEMT group at 63.5 (57–68) days (*p<0.05*; Fig 3).

## Discussion

Oral care has been reported to reduce the risk of OM or minimize its effects. For instance, oral care reduces the severity of OM, shortens the LOD, and reduces the use of morphine in patients treated with chemotherapy [6–9]. Dentists, dental hygienists, and nurses are actively

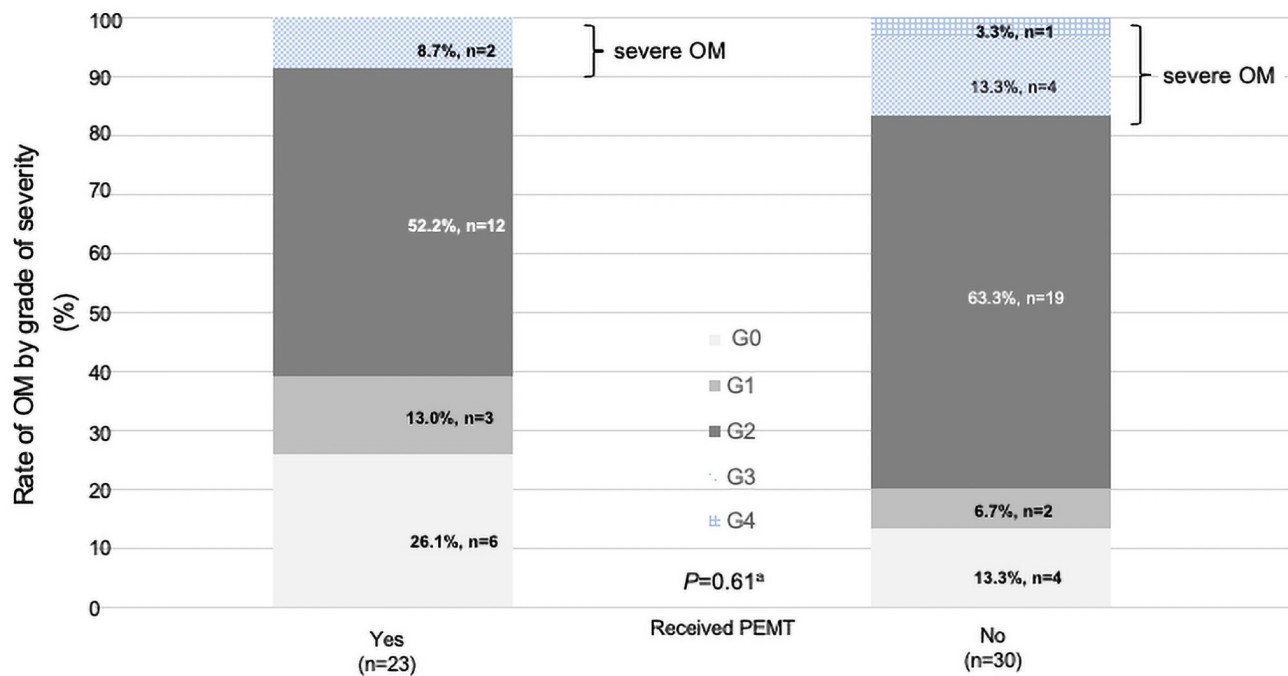

**Fig 2. Severity of oral mucositis by CTCAE grade.** CTCAE = Common terminology criteria for adverse events. [a] Fisher's exact test.

engaged in oral care at many healthcare facilities. The rate of OM has been reported to be 91% in patients with HNC receiving CCRT, and the incidence of Gr 3 and Gr 4 OM—which can determine the need to interrupt or discontinue chemotherapy—has been reported to be 66% [3]. Systematic oral care programs among patients with HNC receiving CCRT are unable to reduce the rate of severe OM but may reduce the infection risk and indirectly improve treatment compliance [13]. In particular, patient education about oral care has been regarded as an important process, and reduction of OM severity has been reported [11]. In addition, randomised controlled study reported that mouth wash and education program promote better physical and quality of life in CCRT [14, 15]. Mouthwash provides a moist environment for the oral mucosa and decreases pain, improves xerostomia and promotes oral comfort [16, 17]. Therefore, it is very important to continue mouthwash properly.

Generally, pharmacists are professionals who are well positioned to explain to patients information about the pharmacological effects of chemotherapy, possible side effects, the timing of side effects, and appropriate drugs for counteracting some of the side effects that may develop. In addition, pharmacists who receive professional oral-care training may further benefit patients. In type 2 diabetes patients, glycated hemoglobin or HbA1c has been shown to be significantly reduced when their pharmacists are highly knowledgeable about diabetes and help them to better manage their condition [18]. Therefore, it may be useful to involve a pharmacist to increase patient adherence to OM management. Patients can benefit from having an oral care team of dentists, dental hygienists, and nurses explain to them how to use a mouthwash and moisturizer. The addition of pharmacists to a medical team provides further support to patients because pharmacists can also explain the necessity of a mouthwash and oral moisturizer to patients. The patient education efforts by pharmacists in this study specifically focused on the frequency of product use and methods of use which were considered to further improve patient adherence to oral care. Our findings showed that PEMT for OM positively affected patients with HNC receiving CCRT.

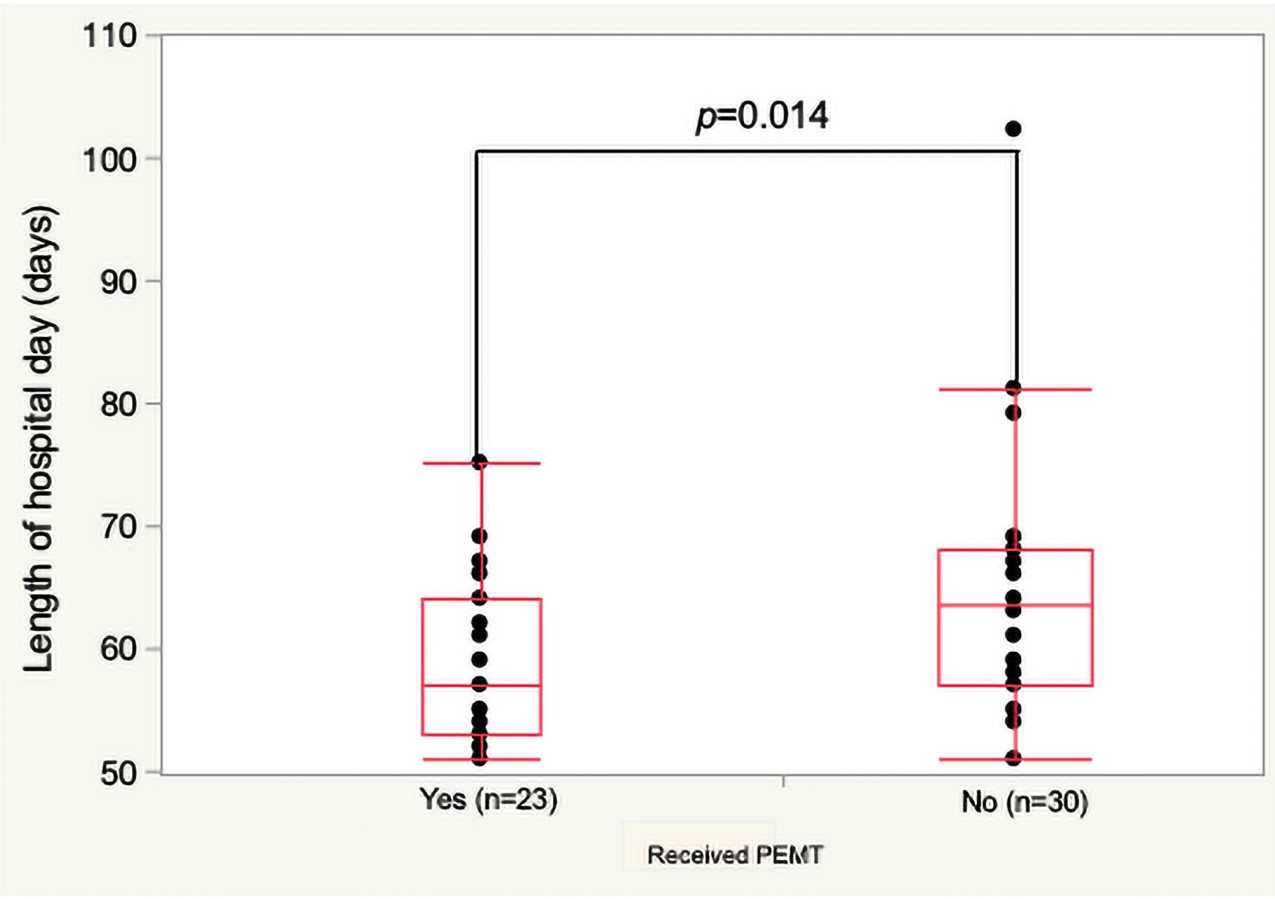

**Fig 3. Length of hospital days.**

The background of the patients assessed in our study revealed that weight loss was significantly less among patients who received PEMT compared with those who did not. A body mass index (BMI) less than 22 kg/m$^2$ has been reported to be a risk factor for moderate-to-severe OM [19]. It was thought that the reduction of weight loss achieved as a consequence of PEMT could limit the risk for OM.

Lidocaine is often used to alleviate pain due to OM. Based on our study results, the pain due to OM was observed to be comparatively less severe among the group of patients that received PEMT. This result was supported by the significantly lower percentage of patients in this group requiring opioids.

In our hospital, opioid was started by oral administration of oxycodone and was switched to fentanyl tape when oral pain made it difficult for the patient to swallow. Switching to fentanyl tape was significantly lower among those patients who received PEMT versus those who did not, possibly because those procedures reduced their oral pain. Enforcement of correct mouthwash and moisturizer use by pharmacists enables the maximal therapeutic advantages to the patients.

There was no difference in the presence or absence of OM and its severity between the PEMT and non-PEMT groups. However, there was a lower incidence of Gr 3 and Gr 4 OM in the PEMT group, although not significantly. For this reason, PEMT was thought to have reduced the adverse events which required judgment on whether CCRT should be interrupted (Fig 2).

LOD was significantly shorter in the PEMT group than the non-PEMT group. PEMT had a significant effect on reduction of weight loss, use of lidocaine viscous, opioid dose, and rate of switch to fentanyl tape. However, we did not perform multivariate analysis because there were not enough events for this analysis.

This study has several limitations. First, the retrospective study may introduce unavoidable bias. Various factors are involved in the pathology of OM, such as oral environment, patient background, and management system of the hospital. The second limitation of our study was the inconsistency in the exact product usage among patients. The usage was challenging to investigate. For example, some cases were required to use a high dose at one time by the prescribing physician, and other cases were already using a high dose prior to hospitalization. Another limitation was that the patients were not surveyed with respect to compliance in this study. Patients with PEMT were asked to verbally confirm that the mouthwash and oral moisturizer were used at least 6 times daily. Each time this regimen was not followed, the pharmacist explained again the need for mouthwash and oral moisturizers. The accuracy of future investigations will likely be improved by the exclusion of the aforementioned OM risk factors from all cases, critical observation of OM cases, and addressing the need for further prospective studies.

## Conclusion

Our results showed that PEMT did not improve OM (yes/no) or OM severity in HNC patients undergoing CCRT. However, we found lower incidence of Gr 3 and Gr 4 OM in the group that received PEMT, although not significantly. In addition, PEMT contributed to oral pain relief and lowered the risk for OM by reduction in weight loss.

## Supporting information

**S1 Dataset.**
(PDF)

## Acknowledgments

We wish to thank Naoto Suzuki, Keiko Tanaka, and Megumi Nakai for organizing the data collection and caring for the study participants.

## Author Contributions

**Conceptualization:** Kensuke Yoshida, Yasumitsu Kodama, Kouji Katsura.

**Data curation:** Akira Toyama.

**Formal analysis:** Kyongsun Pak.

**Investigation:** Yusuke Tanaka, Marie Soga.

**Methodology:** Kouji Katsura.

**Project administration:** Ritsuo Takagi.

**Writing – original draft:** Kensuke Yoshida.

**Writing – review & editing:** Kensuke Yoshida.

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
