## [Decision Letter · Decision Letter 0]

11 Jun 2021

PONE-D-21-15398

Pharmacist involved education program in a multidisciplinary team for oral mucositis: Its impact in head-and-neck cancer patients

PLOS ONE

Dear Dr. Yoshida,

Thank you for submitting your manuscript to PLOS ONE. After careful consideration, we feel that it has merit but does not fully meet PLOS ONE’s publication criteria as it currently stands. Therefore, we invite you to submit a revised version of the manuscript that addresses the points raised during the review process.

We look forward to receiving your revised manuscript.

Kind regards,

Vijayaprakash Suppiah, PhD

Academic Editor

PLOS ONE

Journal Requirements:

2. In your ethics statement in the manuscript and in the online submission form, please ensure that you have discussed whether all data/samples were fully anonymized before you accessed them and/or whether the IRB or ethics committee waived the requirement for informed consent. If patients provided informed written consent to have data/samples from their medical records used in research, please include this information.

3. Please include your tables as part of your main manuscript and remove the individual files. Please note that supplementary tables (should remain/ be uploaded) as separate "supporting information" files

Reviewers' comments:

Reviewer's Responses to Questions

**Comments to the Author**

1. Is the manuscript technically sound, and do the data support the conclusions?

Reviewer #1: Partly

Reviewer #2: Yes

2. Has the statistical analysis been performed appropriately and rigorously? 

Reviewer #1: Yes

Reviewer #2: Yes

3. Have the authors made all data underlying the findings in their manuscript fully available?

Reviewer #1: Yes

Reviewer #2: No

4. Is the manuscript presented in an intelligible fashion and written in standard English?

Reviewer #1: Yes

Reviewer #2: Yes

5. Review Comments to the Author

Reviewer #1: The paper documented the clinical outcomes of a pharmacist-involved education intervention among patients with head and neck cancer receiving concurrent chemoradiotherapy.

Major issues:

- There is potential of selection bias with a retrospective study design. For example, what are the selection criteria for a patient to be enrolled in the intervention compared to those without? The authors need to specify the efforts to reduce this bias. Although baseline characteristics of the two groups showed no statistical significance, but how were the number of study subjects determined and chosen? By inclusion of all and so-happened both groups were equal? Unless it is shown that selection bias is adequately addressed, the results and conclusion of the study may be invalid.

- An in-depth discussion of how a pharmacist helped in improving the clinical outcomes is required. Any documented increase in adherence to the prescribed regimen? It is suggested that more references to previous literatures should be included, especially in the field of oral care to substantiate the authors’ claims.

Minor issues:

- Duplicate keywords. Suggest ‘patient education’ to replace ‘education program’, ‘education’ and ‘advanced dental education’

- Please include statistical analysis in the abstract. Why 53 patients? Total samples during the stipulated timeframe?

- Please include study setting in the abstract

- p-value should not be capitalized

- Table 1: please double check the gender ratio under non-PEMT

- Table 1: weight loss should be an outcome rather than baseline characteristic

- Table 2: total sample was 53 but reduced to 47 in table 2?

- line 2 under introduction: amount

- Please provide the full name of the statistical software under methodology

- What are the criteria for PEMT compared to those without? If without PEMT, does it mean just without pharmacist (other healthcare professionals are available) or without a multidisciplinary team?

- If this was not the patients’ first cycle of chemotherapy, what was the history of OM occurrence between both comparison groups?

- 5th paragraph of discussion: how had the involvement of a pharmacist reduced pain?

Reviewer #2: Notes:

• The number of subjects (53) enrolled was in the result section,not the method section. The 53 subjects were not bad for drawing a conclusion, but a larger sample would be better for future research.

• *A personal preference for the conclusion to be more precise without acronyms (stand-alone)

• For Figure 2:

o The title should have no acronym. (Note: the “MO” better to be written in full). However, It was written on the legend page (page 13), but it was not on the single page of figure 2. Thus, if the final version has the full title, as in the legend, then no issues.

6. PLOS authors have the option to publish the peer review history of their article (what does this mean?). If published, this will include your full peer review and any attached files.

Reviewer #1: **Yes: **Huey Miin Cheah

Reviewer #2: **Yes: **Aliah Aldahash

---

## [Author Response · Author response to Decision Letter 0]

30 Jun 2021

The paper documented the clinical outcomes of a pharmacist-involved education intervention among patients with head and neck cancer receiving concurrent chemoradiotherapy.

Major issues:

- There is potential of selection bias with a retrospective study design. For example, what are the selection criteria for a patient to be enrolled in the intervention compared to those without? The authors need to specify the efforts to reduce this bias. Although baseline characteristics of the two groups showed no statistical significance, but how were the number of study subjects determined and chosen? By inclusion of all and so-happened both groups were equal? Unless it is shown that selection bias is adequately addressed, the results and conclusion of the study may be invalid.

→Thank you very much for your suggestion. Pharmacist was educated by a dentist in January 2018. Therefore, the surveillance period was chosen to compare the year prior to January 2018, when no education was received, and the year after January 2018, when education was received.

We considered that the following as a method to reduce selection bias.

・All patients who underwent CCRT during the investigation period were surveyed as a countermeasure against selection bias. 

・Investigators were not involved CCRT as a countermeasure against medical surveillance bias.

・ Presence/absence of oral mucositis and grade evaluator are not all author. 

It is an exploratory study, and there were no similar reports in the past, so sample size could not be calculated. Therefore, from the viewpoint of feasibility, we investigated two-year survey period and conducted a retrospective survey. With the publication of this paper, it will be possible to calculate the sample size in the future, so I would like to determine patient information in more detail and randomized controlled trials. Thank you for your wonderful suggestions.

We modified the manuscript as follows:

P5L6

This study was performed with the approval of the Research Ethics Niigata university Hospital (approval number 2018-0201). The subjects were patients who were undergoing CCRT from February 2017 to January 2019. All patients received a first cycle of chemotherapy. Pharmacist was educated by a dentist in January 2018. Therefore, the surveillance period was chosen to compare the year prior to January 2018, when no education was received, and the year after January 2018, when education was received. All patients received CCRT on a regimen of cisplatin 80 mg/m2 on days 1, 22, and 43 during the aforementioned time period.

A retrospective survey of electronic medical records was conducted with the cooperation of the Medical Information Department of Niigata university Hospital. The patient data collected included age, gender, weight loss, steroid or immunosuppressant use, hematological values (albumin, white blood cell count, blood platelets, and neutrophils), cancer grade, primary cancer site, type and use of mouthwash and moisturizer, opioid use (yes/no, days until the start of opioid use, and dose), switch to fentanyl tape, and LOD. All patients who underwent CCRT during the investigation period were surveyed as a countermeasure against selection bias. Investigators were not involved CCRT as a countermeasure against medical surveillance bias. Presence/absence of OM and grade evaluator are not all author.

P8L5

Patient background characteristics were summarized by group (PEMT: yes/no). The two groups were compared using Fisher's exact test for qualitative data and the Mann-Whitney U test for quantitative data, and a significance level of p<0.05 was set. JMP® 14.2 (SAS Institute Inc., Cary, NC, USA) was used for statistical analysis. It is an exploratory study, and there were no similar reports in the past, so sample size could not be calculated. Therefore, from the viewpoint of feasibility, we investigated two-year survey period and conducted a retrospective survey.

- An in-depth discussion of how a pharmacist helped in improving the clinical outcomes is required. Any documented increase in adherence to the prescribed regimen? It is suggested that more references to previous literatures should be included, especially in the field of oral care to substantiate the authors’ claims.

- →Thank you very much for your suggestion. This study was retrospective study, so there was no document. Therefore, we investigated increase in adherence by the amount of mouthwash and moisturizer used, but could not follow the detailed information and could not show the statistical difference. However, as the pharmacist explained, we always checked on a face-to-face basis once a week that moisturizing was performed after moisturizer and proper storage of mouthwash (in darkness) was followed. In the future, We would like to proceed with more detailed prospective research. 

We added 4references to previous literatures. Please check again.

P12L14

In addition, randomised controlled study reported that mouth wash and education program promote better physical and quality of life in CCRT［14,15］. Mouthwash provides a moist environment for the oral mucosa and decreases pain, improves xerostomia and promotes oral comfort［16,17］. Therefore, it is very important to continue mouthwash properly.

Minor issues:

- Duplicate keywords. Suggest ‘patient education’ to replace ‘education program’, ‘education’ and ‘advanced dental education’ 

→Thank you for your suggestion. We corrected it.

P5L14　Patient education

- Please include statistical analysis in the abstract. Why 53 patients? Total samples during the stipulated timeframe?

→Thank you for your suggestion. We include statistical analysis in the abstract. 

As you pointed out, there were 53 people during the stipulated timeframe.

We corrected it.

P2L7　

Total samples data of 53 patients during the stipulated timeframe were retrospectively collected from electronic medical records from February 2017 to January 2019. We compared the presence/absence of OM (OM: yes/no) between patients with and without PEMT (PEMT: yes/no) as the primary endpoint and OM severity as the secondary endpoint. 

The following information was surveyed: age, gender, weight loss, steroid or immunosuppressant use, hematological values (albumin, white blood cell count, blood platelets, and neutrophils), cancer grade, primary cancer site, type and use of mouthwash and moisturizer, opioid use (yes/no, days until the start of opioid use, and dose, switch to tape), and length of hospital day (LOD). The two groups were compared using Fisher's exact test for qualitative data and the Mann-Whitney U test for quantitative data, and a significance level of p<0.05 was set.

- Please include study setting in the abstract

→Thank you for your suggestion. We include study setting in the abstract. Same as the above correction.

- p-value should not be capitalized

→We corrected it.

- Table 1: please double check the gender ratio under non-PEMT

→Thank you very much for your suggestion. We corrected it. Please check again.

- Table 1: weight loss should be an outcome rather than baseline characteristic

→Thank you very much for your suggestion. We added Table. 

Table 2. Weight loss

- Table 2: total sample was 53 but reduced to 47 in table 2?

→Thank you very much for your suggestion. Because patients who have already used opioids before the target period are not included summary of data for patients receiving opioids. Very sorry for the lack of explanation. We added manuscript.

P6L6

The opioid dose used was determined from the oral morphine conversion ratio of the Japanese Society for Palliative Medicine as follows: oral morphine: oral oxycodone: transdermal fentanyl = (30:20:1) ［12］. Patients who have already used opioids before the target period are not included summary of data for patients receiving opioids. The LOD was the period from the CCRT start date to discharge from the hospital.

- line 2 under introduction: amount

→Thank you very much for your suggestion. We corrected it. Please check again.

P4L3

Oral mucositis (OM) has been described as the most painful aspect of chemotherapy［1］. Due to increased pain attributed to OM, the ingested amount of food is decreased, motivation with respect to treatment is reduced, and treatment-related deaths can occur, all of which warrant a change in the therapeutic approach［2］.

- Please provide the full name of the statistical software under methodology

→Thank you for your suggestion. We corrected JMP® 14.2 (SAS Institute Inc., Cary, NC, USA).

P8L5

Patient background characteristics were summarized by group (PEMT: yes/no). The two groups were compared using Fisher's exact test for qualitative data and the Mann-Whitney U test for quantitative data, and a significance level of p<0.05 was set. JMP® 14.2 (SAS Institute Inc., Cary, NC, USA) was used for statistical analysis.

- What are the criteria for PEMT compared to those without? If without PEMT, does it mean just without pharmacist (other healthcare professionals are available) or without a multidisciplinary team?

→Thank you very much for your suggestion. Pharmacist was educated by a dentist in January 2018. Therefore, the surveillance period was chosen to compare the year prior to January 2018, when no education was received, and the year after January 2018, when education was received. →Same as the above correction.

If without PEMT, it mean that pharmacist did not educate CCRT patient (other healthcare professionals are available).

- If this was not the patients’ first cycle of chemotherapy, what was the history of OM occurrence between both comparison groups?

→Thank you for your suggestion. All patients received a first cycle of chemotherapy. We added manuscript.

P5L6

This study was performed with the approval of the Research Ethics Niigata university Hospital (approval number 2018-0201). The subjects were patients who were undergoing CCRT from February 2017 to January 2019. All patients received a first cycle of chemotherapy.

- 5th paragraph of discussion: how had the involvement of a pharmacist reduced pain?

→Thank you for your suggestion. The involvement of a pharmacist reduced lidocaine viscous, opioid doses, and the rate of fentanyl tape switching due to mouth pain. We considered that this result was attributed to the correct use of mouthwash and moisturizer to reduce the severity of OM.

P14L5

PEMT had a significant effect on reduction of weight loss, use of lidocaine viscous, opioid dose, and rate of switch to fentanyl tape.

Other

We corrected the manuscript according to the PLOS ONE guidelines. Please check again.

About the supplementary information, we did not see any need for it.

We would like to appreciate it if you could tell us what information you need.

---

## [Decision Letter · Decision Letter 1]

22 Jul 2021

PONE-D-21-15398R1

Pharmacist involved education program in a multidisciplinary team for oral mucositis: Its impact in head-and-neck cancer patients

PLOS ONE

Dear Dr. Yoshida,

Thank you for submitting your manuscript to PLOS ONE. After careful consideration, we feel that it has merit but does not fully meet PLOS ONE’s publication criteria as it currently stands. Therefore, we invite you to submit a revised version of the manuscript that addresses the points raised during the review process.

We look forward to receiving your revised manuscript.

Kind regards,

Vijayaprakash Suppiah, PhD

Academic Editor

PLOS ONE

Journal Requirements:

Reviewers' comments:

Reviewer's Responses to Questions

**Comments to the Author**

1. If the authors have adequately addressed your comments raised in a previous round of review and you feel that this manuscript is now acceptable for publication, you may indicate that here to bypass the “Comments to the Author” section, enter your conflict of interest statement in the “Confidential to Editor” section, and submit your "Accept" recommendation.

Reviewer #1: All comments have been addressed

Reviewer #2: All comments have been addressed

2. Is the manuscript technically sound, and do the data support the conclusions?

Reviewer #1: Yes

Reviewer #2: Yes

3. Has the statistical analysis been performed appropriately and rigorously? 

Reviewer #1: Yes

Reviewer #2: Yes

4. Have the authors made all data underlying the findings in their manuscript fully available?

Reviewer #1: Yes

Reviewer #2: Yes

5. Is the manuscript presented in an intelligible fashion and written in standard English?

Reviewer #1: Yes

Reviewer #2: Yes

6. Review Comments to the Author

Reviewer #1: Thank you very much for the efforts in the latest revision. The article is now more convincing and reads better.

Just a few minor suggestions as below:

1. Please double check the affiliation superscripts on the title page. What’s the difference between:

a. ¶: These authors contributed equally to this work.

b. &: These authors also contributed equally to this work.

2. Please ensure p-value is not capitalised in the main text and the figures

3. Please standardize to use either Figure or Fig.

4. Materials and methods> Subjects>

a. Please capitalise ‘University Hospital’

b. ‘the’ first cycle of chemotherapy

c. >Definition of terms: please either define RT or use CCRT

5. Results> Opioids> line 2 should be (Table 3)

6. There are differences between values in Table 3 and in the result paragraph. Please double check.

7. Definition of RT is not needed in Table 1 since the term isn’t there.

8. Please remove the unnecessary ‘ ) ’ after a superscript annotation in all tables

9. Maybe can consider to the delete the intepretation of whisker plot?

a. Fig 3 Length of hospital days. a Data are shown in a box and whisker plot. The ends of the box represent the upper and lower quartiles and the median is depicted by the horizontal line within the box.

Reviewer #2: The research is original, it has not been published nor the same data and analysis used elsewhere. However, the statistical analysis in the methodology was clarified. Also, the authors provide approval from their local area where the study was conducted.

In terms of the previous feedback, the authors have addressed the comments.

7. PLOS authors have the option to publish the peer review history of their article (what does this mean?). If published, this will include your full peer review and any attached files.

Reviewer #1: No

Reviewer #2: **Yes: **Aliah Aldahash

---

## [Author Response · Author response to Decision Letter 1]

28 Jul 2021

Thank you very much for the efforts in the latest revision. The article is now more convincing and reads better.

Just a few minor suggestions as below: 

Thank you very much for carefully checking our paper. We were able to correct it to a better manuscript.

1. Please double check the affiliation superscripts on the title page. What’s the difference between:

a. ¶: These authors contributed equally to this work.

b. &: These authors also contributed equally to this work.

→Thank you very much for your suggestion. According to PLOS ONE's style requirements, we marked ¶ for the firstly contributed equally to this work and & for the secondly contributed equally to this work.

P1L21

¶: These authors contributed firstly equally to this work.

&: These authors also contributed secondly equally to this work.

2. Please ensure p-value is not capitalised in the main text and the figures

→Thank you very much for your suggestion. We corrected it.

P8L19

Of the 53 patients enrolled in this study, 23 patients received PEMT and 30 patients did not. The combination of SASH + sodium bicarbonate (NaHCO3) + lidocaine hydrochloride viscous was used by 0% of the patients who received PEMT versus 30% (9 patients; p<0.01 Table 1) of those who did not.

3. Please standardize to use either Figure or Fig.

→Thank you very much for your suggestion. We standardize to use Fig.

Fig1. Pharmacist-involved education program in the multidisciplinary team 

4. Materials and methods> Subjects> 

a. Please capitalise ‘University Hospital’

b. ‘the’ first cycle of chemotherapy

c. >Definition of terms: please either define RT or use CCRT

→Thank you very much for your suggestion. We corrected it.

P5L6

This study was performed with the approval of the Research Ethics Niigata University Hospital (approval number 2018-0201). All data were fully anonymized, so the Research Ethics Niigata University Hospital waived the requirement for informed consent.

P5L17

A retrospective survey of electronic medical records was conducted with the cooperation of the Medical Information Department of Niigata University Hospital.

P5L9

All patients received the first cycle of chemotherapy. Pharmacist was educated by a dentist in January 2018.

P6L5

Weight loss was defined as the difference in body weight before the initiation of CCRT (0 Gy) and after the delivery of all fractions (70 Gy).

5. Results> Opioids> line 2 should be (Table 3)

→Thank you very much for your suggestion. We corrected it.

P10L5

Opioids were used by 65.2% of patients (15 cases) who received PEMT and by 92.0% (23 cases) of those who did not; although the difference between groups was not significant (Table 3).

6. There are differences between values in Table 3 and in the result paragraph. Please double check.

→Thank you very much for your suggestion. We corrected it.

P10L9

The median opioid dose was significantly different between groups; 1102.5 (0–1736.5) mg in the PEMT group and 1860.5 (1050.5–2850.0) mg in the non-PEMT group (p<0.01).

7. Definition of RT is not needed in Table 1 since the term isn’t there.

→Thank you very much for your suggestion. We deleted RT.

8. Please remove the unnecessary ‘ ) ’ after a superscript annotation in all tables

→Thank you very much for your suggestion. We deleted its, please check again.

9. Maybe can consider to the delete the intepretation of whisker plot? 

a. Fig 3 Length of hospital days. a Data are shown in a box and whisker plot. The ends of the box represent the upper and lower quartiles and the median is depicted by the horizontal line within the box.

→Thank you very much for your suggestion. We deleted it.

P12L1

Fig 3 Length of hospital days. 

Other

We corrected references.

P16L1

1. Bellm LA, Epstein JB, Rose-Ped A, Martin P, Fuchs HJ. Patient reports of complications of bone marrow transplantation. Support Care Cancer. 2000; 8: 33–39.

2. Keefe DM, Schubert MM, Elting LS, Sonis ST, Epstein JB, Raber-Durlacher JE, et al. Mucositis Study Section of the Multinational Association of Supportive Care in Cancer and the International Society for Oral Oncology. Updated clinical practice guidelines for the prevention and treatment of mucositis. Cancer 2007; 109: 820–831.

3. Elting LS, Cooksley CD, Chambers MS, Garden AS. Risk, outcomes, and costs of radiation-induced oral mucositis among patients with head-and-neck malignancies. Int J Radiat Oncol Biol Phys. 2007; 68: 1110–1120.

4. Peterson DE, Boers-Doets CB, Bensadoun RJ, Herrstedt J, ESMO Guidelines Committee. Management of oral and gastrointestinal mucosal injury: ESMO Clinical Practice Guidelines for diagnosis, treatment, and follow-up. Ann Oncol. 2015; 5: 139–151.

5. Elad S, Cheng KKF, Lalla RV, Yarom N, Hong C, Logan RM, et al. Mucositis Guidelines Leadership Group of the Multinational Association of Supportive Care in Cancer and International Society of Oral Oncology (MASCC/ISOO)(2020) MASCC/ISOO clinical practice guidelines for the management of mucositis secondary to cancer therapy. Cancer. 2020; 126: 4423–4431.

6. Saito H, Watanabe Y, Sato K, Ikawa H, Yoshida Y, Katakura A, et al. Effects of professional oral health care on reducing the risk of chemotherapy-induced oral mucositis. Support Care Cancer. 2014; 22: 2935–2940.

7. Kubota K, Kobayashi W, Sasaki H, Nakagawa H, Kon T, Mimura M, et al. Professional oral health care reduces oral mucositis pain in patients treated by superselective intra-arterial chemotherapy concurrent with radiotherapy for oral cancer. Support Care Cancer. 2015; 23: 3323–3329.

8. Legert KG, Remberger M, Ringden O, Heimdahl A, Dahllof G. Reduced intensity conditioning and oral care measures prevent oral mucositis and reduces days of hospitalization in allogeneic stem cell transplantation recipients. Support Care Cancer. 2014; 22: 2133–2140.

9. Soga Y, Sugiura Y, Takahashi K, Nishimoto H, Maeda Y, Tanimoto M, et al. Progress of oral care and reduction of oral mucositis--a pilot study in a hematopoietic stem cell transplantation ward. Support Care Cancer. 2010; 19: 303–307.

10. The European Oral Care in Cancer Group (EOCC). Oral care guidance 2018. Japanese Association of Supportive Care in Cancer Web site. http://jascc.jp/wp/wp-content/uploads/2018/01/guidance_20180115.pdf. Accessed February 20, 2020. 

11. Leppla L, Geest SD, Fierz K, Deschler-Baier B, Koller A. An oral care self-management support protocol (OrCaSS) to reduce oral mucositis in hospitalized patients with acute myeloid leukemia and allogeneic hematopoietic stem cell transplantation: a randomized controlled pilot study. Support Care Cancer. 2016; 24: 773–782.

12. Japanese Society for Palliative Medicine. Japanese Society for Palliative Medicine Guidelines 2010. Japanese Society for Palliative Medicine Web site. www.jspm.ne.jp/guidelines/pain/2010/chapter02/02 04 01 05.php. Accessed February 20, 2020. 

13. Yokota T, Tachibana H, Konishi T, Yurikusa T, Hamauchi S, Sakai K, et al. Multicenter phase II study of an oral care program for patients with head and neck cancer receiving chemoradiotherapy. Support Care Cancer. 2016; 24: 3029–3036.

14. Yüce UÖ, Yurtsever S. Effect of Education About Oral Mucositis Given to the Cancer Patients Having Chemotherapy on Life Quality. J Cancer Educ. 2019; 34: 35-40.

15. B-S Huang, S-C Wu, C-Y Lin, K-H Fan, J T-C Chang, S-C Chen. The effectiveness of a saline mouth rinse regimen and education programme on radiation-induced oral mucositis and quality of life in oral cavity cancer patients: A randomised controlled trial. Eur J Cancer Care (Engl). 2018; 27: e12819.

16. Dorothy M Keefe, Mark M Schubert, Linda S Elting, Stephen T Sonis, Joel B Epstein, Judith E Raber-Durlacher et al. Updated clinical practice guidelines for the prevention and treatment of mucositis. Cancer. 2007; 109: 820-831.

17. Edward B Rubenstein, Douglas E Peterson, Mark Schubert, Dorothy Keefe, Deborah McGuire, Joel Epstein. Clinical practice guidelines for the prevention and treatment of cancer therapy-induced oral and gastrointestinal mucositis. Cancer. 2004; 100: 2026-2046.

18. Okada H, Onda, Shoji, Kotani K, Nakayama T, Nakagawa Y, Sakane N. Effects of lifestyle intervention performed by community pharmacists on glycemic control in patients with type 2 diabetes: the community pharmacists assist (Compass) project, a pragmatic cluster randomized trial. Pharmacol Pharmacy. 2016; 7: 124–132.

19. Saito N, Imai Y, Muto T, Sairenchi T. Low body mass index as a risk factor of moderate to severe oral mucositis in oral cancer patients with radiotherapy. Support Care Cancer. 2012; 20: 3373–3377.

---

## [Decision Letter · Decision Letter 2]

25 Aug 2021

PONE-D-21-15398R2

Pharmacist involved education program in a multidisciplinary team for oral mucositis: Its impact in head-and-neck cancer patients

PLOS ONE

Dear Dr. Yoshida,

Thank you for submitting your manuscript to PLOS ONE. After careful consideration, we feel that it has merit but does not fully meet PLOS ONE’s publication criteria as it currently stands. Therefore, we invite you to submit a revised version of the manuscript that addresses the points raised during the review process.

We look forward to receiving your revised manuscript.

Kind regards,

Rohit Kunnath Menon

Academic Editor

PLOS ONE

Journal Requirements:

Reviewers' comments:

Reviewer's Responses to Questions

**Comments to the Author**

1. If the authors have adequately addressed your comments raised in a previous round of review and you feel that this manuscript is now acceptable for publication, you may indicate that here to bypass the “Comments to the Author” section, enter your conflict of interest statement in the “Confidential to Editor” section, and submit your "Accept" recommendation.

Reviewer #1: (No Response)

Reviewer #2: All comments have been addressed

2. Is the manuscript technically sound, and do the data support the conclusions?

Reviewer #1: Yes

Reviewer #2: Yes

3. Has the statistical analysis been performed appropriately and rigorously? 

Reviewer #1: Yes

Reviewer #2: Yes

4. Have the authors made all data underlying the findings in their manuscript fully available?

Reviewer #1: Yes

Reviewer #2: Yes

5. Is the manuscript presented in an intelligible fashion and written in standard English?

Reviewer #1: Yes

Reviewer #2: Yes

6. Review Comments to the Author

Reviewer #1: Table 2 and Table 3 > please delete the ‘)’ behind a superscript

Page 10 and Table 3, Opioids> consider to delete the trailing zeros, 19 and 18

Table 3> Maybe need to provide a brief reason for the exclusion at footnote a and b (eg because patient(s) was/were on opioids before CCRT)

The figures in Table 3 and result paragraph ‘Opioids’ are still different. 1860.5 in paragraph but 1860.0 in Table 3. 1736.5 in paragraph but 1736.3 in Table 3. Consider to remove all trailing zeros (eg 15, instead of 15.0)

Please standardize references to use small caps for all the first alphabets of each word in the journal article/report title.

Discussion> Switching to fentanyl tape was significantly lower among those patients who received PEMT versus those who did not, possibly because those procedures reduced their oral pain.> Consider to elaborate this (eg. Enforcement of correct mouthwash and moisturizer use by pharmacists enables the maximal therapeutic advantages to the patients)

Reviewer #2: The research is original, it has not been published nor the same data and analysis used elsewhere. The methodology and the statistical analysis were clarified. Also, the authors provide approval from their local area where the study was conducted.

In terms of the previous feedback, the authors have addressed the comments.

7. PLOS authors have the option to publish the peer review history of their article (what does this mean?). If published, this will include your full peer review and any attached files.

Reviewer #1: **Yes: **Cheah Huey Miin

Reviewer #2: **Yes: **Aliah Aldahash

---

## [Author Response · Author response to Decision Letter 2]

29 Aug 2021

Reviewer #1: Table 2 and Table 3 > please delete the ‘)’ behind a superscript

→Thank you very much for your suggestion. We deleted its, please check again.

Page 10 and Table 3, Opioids> consider to delete the trailing zeros, 19 and 18

→Thank you very much for your suggestion. We deleted its, please check again.

P10

Opioids were used by 65.2% of patients (15 cases) who received PEMT and by 92% (23 cases) of those who did not; although the difference between groups was not significant (Table 3). Days until the start of opioid use were 19 (17–21.5) days in the PEMT group and 18 (15–21.5) days in the non-PEMT group, resulting in no significant difference between groups. The median opioid dose was significantly different between groups; 1102.5 (0–1736.3) mg in the PEMT group and 1860 (1050–2850) mg in the non-PEMT group (p<0.01). Oral oxycodone was switched to fentanyl tape in a significantly lower percentage of patients in the PEMT group (22.7%; 5 cases) than non-PEMT group (64.0%; 16 cases) (Table 3).

Table 3> Maybe need to provide a brief reason for the exclusion at footnote a and b (eg because patient(s) was/were on opioids before CCRT)

→Thank you very much for your suggestion. We added it, please check again.

The figures in Table 3 and result paragraph ‘Opioids’ are still different. 1860.5 in paragraph but 1860.0 in Table 3. 1736.5 in paragraph but 1736.3 in Table 3. Consider to remove all trailing zeros (eg 15, instead of 15.0)

→Thank you very much for checking carefully. We corrected its, please check again. Same as the above correction.

Please standardize references to use small caps for all the first alphabets of each word in the journal article/report title.

→Thank you very much for checking carefully. We corrected its.

1. Bellm LA, Epstein JB, Rose-Ped A, Martin P, Fuchs HJ. Patient reports of complications of bone marrow transplantation. Support Care Cancer. 2000; 8: 33–39.

2. Keefe DM, Schubert MM, Elting LS, Sonis ST, Epstein JB, Raber-Durlacher JE, et al. Mucositis Study Section of the Multinational Association of Supportive Care in Cancer and the International Society for Oral Oncology. Updated clinical practice guidelines for the prevention and treatment of mucositis. Cancer 2007; 109: 820–831.

3. Elting LS, Cooksley CD, Chambers MS, Garden AS. Risk, outcomes, and costs of radiation-induced oral mucositis among patients with head-and-neck malignancies. Int J Radiat Oncol Biol Phys. 2007; 68: 1110–1120.

4. Peterson DE, Boers-Doets CB, Bensadoun RJ, Herrstedt J, ESMO Guidelines Committee. Management of oral and gastrointestinal mucosal injury: ESMO Clinical Practice Guidelines for diagnosis, treatment, and follow-up. Ann Oncol. 2015; 5: 139–151.

5. Elad S, Cheng KKF, Lalla RV, Yarom N, Hong C, Logan RM, et al. Mucositis Guidelines Leadership Group of the Multinational Association of Supportive Care in Cancer and International Society of Oral Oncology (MASCC/ISOO)(2020) MASCC/ISOO clinical practice guidelines for the management of mucositis secondary to cancer therapy. Cancer. 2020; 126: 4423–4431.

6. Saito H, Watanabe Y, Sato K, Ikawa H, Yoshida Y, Katakura A, et al. Effects of professional oral health care on reducing the risk of chemotherapy-induced oral mucositis. Support Care Cancer. 2014; 22: 2935–2940.

7. Kubota K, Kobayashi W, Sasaki H, Nakagawa H, Kon T, Mimura M, et al. Professional oral health care reduces oral mucositis pain in patients treated by superselective intra-arterial chemotherapy concurrent with radiotherapy for oral cancer. Support Care Cancer. 2015; 23: 3323–3329.

8. Legert KG, Remberger M, Ringden O, Heimdahl A, Dahllof G. Reduced intensity conditioning and oral care measures prevent oral mucositis and reduces days of hospitalization in allogeneic stem cell transplantation recipients. Support Care Cancer. 2014; 22: 2133–2140.

9. Soga Y, Sugiura Y, Takahashi K, Nishimoto H, Maeda Y, Tanimoto M, et al. Progress of oral care and reduction of oral mucositis-a pilot study in a hematopoietic stem cell transplantation ward. Support Care Cancer. 2010; 19: 303–307.

10. The European Oral Care in Cancer Group (EOCC). Oral care guidance 2018. Japanese Association of Supportive Care in Cancer Web site. http://jascc.jp/wp/wp-content/uploads/2018/01/guidance_20180115.pdf. Accessed February 20, 2020. 

11. Leppla L, Geest SD, Fierz K, Deschler-Baier B, Koller A. An oral care self-management support protocol (OrCaSS) to reduce oral mucositis in hospitalized patients with acute myeloid leukemia and allogeneic hematopoietic stem cell transplantation: a randomized controlled pilot study. Support Care Cancer. 2016; 24: 773–782.

12. Japanese Society for Palliative Medicine. Japanese Society for Palliative Medicine Guidelines 2010. Japanese Society for Palliative Medicine Web site. www.jspm.ne.jp/guidelines/pain/2010/chapter02/02 04 01 05.php. Accessed February 20, 2020. 

13. Yokota T, Tachibana H, Konishi T, Yurikusa T, Hamauchi S, Sakai K, et al. Multicenter phase II study of an oral care program for patients with head and neck cancer receiving chemoradiotherapy. Support Care Cancer. 2016; 24: 3029–3036.

14. Yüce UÖ, Yurtsever S. Effect of Education About Oral Mucositis Given to the Cancer Patients Having Chemotherapy on Life Quality. J Cancer Educ. 2019; 34: 35-40.

15. B-S Huang, S-C Wu, C-Y Lin, K-H Fan, J T-C Chang, S-C Chen. The effectiveness of a saline mouth rinse regimen and education programme on radiation-induced oral mucositis and quality of life in oral cavity cancer patients: A randomised controlled trial. Eur J Cancer Care (Engl). 2018; 27: e12819.

16. Dorothy M Keefe, Mark M Schubert, Linda S Elting, Stephen T Sonis, Joel B Epstein, Judith E Raber-Durlacher et al. Updated clinical practice guidelines for the prevention and treatment of mucositis. Cancer. 2007; 109: 820-831.

17. Edward B Rubenstein, Douglas E Peterson, Mark Schubert, Dorothy Keefe, Deborah McGuire, Joel Epstein. Clinical practice guidelines for the prevention and treatment of cancer therapy-induced oral and gastrointestinal mucositis. Cancer. 2004; 100: 2026-2046.

18. Okada H, Onda, Shoji, Kotani K, Nakayama T, Nakagawa Y, Sakane N. Effects of lifestyle intervention performed by community pharmacists on glycemic control in patients with type 2 diabetes: the community pharmacists assist (Compass) project, a pragmatic cluster randomized trial. Pharmacol Pharmacy. 2016; 7: 124–132.

19. Saito N, Imai Y, Muto T, Sairenchi T. Low body mass index as a risk factor of moderate to severe oral mucositis in oral cancer patients with radiotherapy. Support Care Cancer. 2012; 20: 3373–3377.

Discussion> Switching to fentanyl tape was significantly lower among those patients who received PEMT versus those who did not, possibly because those procedures reduced their oral pain.> Consider to elaborate this (eg. Enforcement of correct mouthwash and moisturizer use by pharmacists enables the maximal therapeutic advantages to the patients)

→Thank you very much for your suggestion. We added it, please check again.

P13L23

In our hospital, opioid was started by oral administration of oxycodone and was switched to fentanyl tape when oral pain made it difficult for the patient to swallow. Switching to fentanyl tape was significantly lower among those patients who received PEMT versus those who did not, possibly because those procedures reduced their oral pain. Enforcement of correct mouthwash and moisturizer use by pharmacists enables the maximal therapeutic advantages to the patients.

Reviewer #2: The research is original, it has not been published nor the same data and analysis used elsewhere. The methodology and the statistical analysis were clarified. Also, the authors provide approval from their local area where the study was conducted.

In terms of the previous feedback, the authors have addressed the comments.

---

## [Editor Report · Decision Letter 3]

2 Nov 2021

Pharmacist involved education program in a multidisciplinary team for oral mucositis: Its impact in head-and-neck cancer patients

PONE-D-21-15398R3

Dear Dr. Yoshida,

We’re pleased to inform you that your manuscript has been judged scientifically suitable for publication and will be formally accepted for publication once it meets all outstanding technical requirements.

Kind regards,

Rohit Kunnath Menon

Academic Editor

PLOS ONE
---

## [Editor Report · Acceptance letter]

10 Nov 2021

PONE-D-21-15398R3 

*Pharmacist involved education program in a multidisciplinary team for oral mucositis: its impact in head-and-neck cancer patients*

Dear Dr. Yoshida:

I'm pleased to inform you that your manuscript has been deemed suitable for publication in PLOS ONE. Congratulations! Your manuscript is now with our production department. 

Kind regards, 

on behalf of

Dr. Rohit Kunnath Menon 

Academic Editor

PLOS ONE